# Exploratory Data Analysis for the Evaluation of Tribological Properties of Wear-Resistant Surface Layers Modified with Rare-Earth Metals

**DOI:** 10.3390/ma15062032

**Published:** 2022-03-09

**Authors:** Paweł Malinowski, Justyna Kasińska, Sławomir Rutkowski, Monika Madej

**Affiliations:** 1Faculty of Foundry Engineering, AGH University of Science and Technology, 30-059 Krakow, Poland; pamalino@agh.edu.pl; 2Department of Metal Science and Materials Technology, Kielce University of Technology, 25-314 Kielce, Poland; mmadej@tu.kielce.pl; 3The Faculty of Mechanics and Technology, Rzeszow University of Technology, 37-450 Stalowa Wola, Poland; s.rutkowski@prz.edu.pl; 4Multi-Branch Company T.S.A. Marcin Górski Sławomir Rutkowski, 37-450 Stalowa Wola, Poland

**Keywords:** EDA—exploratory data analysis, rare Earth metal oxides, wear resistance, heatmap, correlation

## Abstract

The role of rare Earth metals in the improvement of the properties of metals and alloys has been analysed and described in multiple studies. Their effects on changes in microstructure and mechanical properties are most pronounced. This paper focuses on the beneficial effect of rare Earth metal oxides on the wear resistance of surface layers applied to castings intended for structural elements of machinery and equipment in mining and recycling. The experiment involved modifying prepared surfaces by adding CeO_2_, Y_2_O_3_, and La_2_O_3_. Hardness measurements, a scratch test, and tribological tests were performed under dry and fluid friction. The maximum wear track depth and track area were measured from the surface profile. To determine correlations between the results, exploratory data analysis was employed. Heatmaps were used to illustrate strong positive and negative interactions. The addition of oxides at increasing carbon content resulted in increased hardness, lower coefficient of friction, and reduced track area and maximum track depth. Strong negative interactions between the track area and maximum track depth were found. The differences resulting from the test conditions (fluid and dry friction) were discussed. This study demonstrated the suitability of exploratory data analysis for analysing research results and confirmed the improvement of modified surface wear resistance.

## 1. Introduction

The interest in rare Earth elements observed in recent years is driven by rising application opportunities provided by new electronic technologies and nanotechnologies [1,2,3,4]. In addition to these specialized areas of the economy, rare Earth metals (REM) are still used in the metallurgical, chemical, and metal industries. Because of their physical and chemical properties, rare Earth oxides began to be used in surface engineering. Their beneficial effect of reducing secondary dendrites and volume fraction of non-metallic inclusions was noticed in the case of nickel-based [5,6] or iron-based [7] coating microstructures. Another advantage was the increased corrosion resistance and improved passivation. In their work [8], Liu et al. reported an advantageous effect of yttrium oxide Y_2_O_3_ on the modification of the nickel-based coating, indicating, however, that the range of its application is limited (from 0.4 to 0.6%).

By influencing changes in the microstructure of steel, alloys, and metallic layers, REM directly improve tribological properties, corrosion resistance, mechanical properties, and resistance to oxidation [9,10,11]. Rare Earth metal additions change the parameters of the structure, e.g., grain size, number and size of inclusions [12,13,14,15], and the mechanical. Adding La_2_O_3_, Gd_2_O_3_, Lu_2_O_3_ particles increases the hardness of surface layers [16,17]. The authors of the work [18] showed that in the case of Si_3_N_4_ ceramics, the addition of rare Earth oxides reduced the friction coefficient and wear. Silicon nitride ceramics sintered with the addition of rare Earth oxides also constitute an important class of materials for high temperature applications. In addition to high temperature strengths, they possess good thermal shock resistance, creep resistance, and high oxidation resistance [16,19,20,21,22].

A number of tests and devices can be applied to determine tribological properties of materials and obtain a broad spectrum of information [23,24,25]. Using only selected parameters, e.g., a friction coefficient or linear wear, and ignoring mass loss, for example, a proper description of the tested material is impossible. In addition, the results obtained are affected by the test conditions, e.g., temperature, friction pair, motion, medium—dry friction and fluid friction [26].

Data analysis or elements of artificial intelligence, therefore, are increasingly used to correctly interpret scientific results. Exploratory data analysis was introduced to support scientific processes and use statistical methods in solving real problems. Exploratory analysis allows for determining the correlation between the results, its type and strength, as well as exclude or significantly limit the influence of the human factor [27,28].

Exploratory data analysis (EDA) may be defined as the art of looking at one or more datasets in an effort to understand the underlying structure of the data contained there [29,30].

The current paper uses the methods of exploratory data analysis to show the relationships between the selected properties of the layers. The authors also wanted to demonstrate the suitability of EDA for analysing the results of tribological tests.

## 2. Methodology

### 2.1. Surface Modification

As part of the experiment, the weld deposit was prepared using metallic powders and rare Earth oxides (CeO_2_, Y_2_O_3_, La_2_O_3_) (Pol-Aura, Olsztyn, Poland). Nickel-based powders (COB-ARC, Chorzow, Poland), iron-based powders (COB-ARC, Chorzow, Poland), and powders containing chromium and tungsten carbides (COB-ARC, Chorzow, Poland) were used to produce the deposits. The weld metals were then applied on S355 steel (Hut-Trans, Katowice, Poland) using metal active gas (MAG) welding in combination with plasma arc welding (PAW). Metallographic specimens were prepared from the collected samples for microscopic observation of microstructural changes. For this purpose, a Phenom XL scanning electron microscope (Thermo Fisher Scientific /Phenom-World, Eindhoven, The Netherlands) was employed. Representative images of the pads made of metallic powder on a nickel matrix (chemical composition: 0.6% C, 3% Fe, 11% Cr, 3.8% Si, the rest Ni) are shown in Figure 1. The modification changed the morphology and arrangement of dendrites, which was reflected in the change of wear resistance.

### 2.2. Tribological Tests

Tribological tests under dry friction and fluid friction with the addition of SiO_2_ were carried out for the prepared variants, in which tribological characteristics (friction coefficient, linear wear) and wear indices (maximum wear track depth, track area) were determined.

The following tests were performed for all the welds: hardness test:
-micro combi tester MCT^3^ ANTON PAAR (Anton Paar, Corcelles-Cormondreche, Switzerland),-nominal loading force 2000 mN,-loading/unloading rate = 4000 mN/min,-Vickers indenter.scratch test
-micro combi tester MCT^3^ ANTON PAAR (Anton Paar, Corcelles-Cormondreche, Switzerland),-initial loading force 30 mN, -final loading force 15,000 mN-loading/unloading rate = 59,979.8 mN/min,-Rockwell indenter-indenter radius 100 μm.tribological tests under dry and fluid friction:
-tribometer TRB3 Anton Paar (Anton Paar, Corcelles-Cormondreche, Switzerland),-reciprocating motion,-amplitude: 10 mm-frequency: 1 Hz-number of cycles: 10,000-friction type: dry friction/fluid friction (water solution 10% SiO_2_)-temperature: 23 ± 1°-humidity: 50 ± 1%tests for the maximum wear track depth and area measured from the surface profile:
-optical profilometer: Leica DCM8 (Leica, Wetzlar, Germany)-the maximum track depth and the track area measured from the surface profile were taken as measures of the sample wear.

The tribological tests allowed determining wear resistance of the welds. Table 1 shows sample results from the experiments performed. 

### 2.3. Exploratory Data Analysis

The data analysis was performed in three steps:

Acquisition

Wrangling

Exploration

In the first stage, exploratory data analysis (EDA) was carried out, which included description and visualisation of the data without assuming any initial hypotheses. The description and visualisation of the data made it possible to identify trends, patterns, missing data, outliers, etc. 

Prior to exploratory data analysis, the following questions were generated:What are your analysis goals and outcomes?What tasks do you perform during analysis?What tools do you use?

The first thing to do with any data set is to read it. This is done not only to get to know all the data collected, but also to reduce the workload during analysis. The initial data investigation is known as exploratory data analysis or EDA and it primarily focuses on visually inspecting the data. The main aim of EDA is to understand what data you have, what possible trends there are, and therefore which statistical tests will be appropriate to use [28].

In the EDA process, descriptive and visualisation analyses were performed, including data set description (number of samples, number of not e number ( NaN )values), the removal of columns with a large number of empty NaNs, the insertion of missing data using strategy = mean, descriptive analysis (mean, standard deviation, min, max, median, 1st quartile, 3rd quartile), the visualisation of elements in individual classes, the identification of outliers, unsupervised learning using clustering, heat maps showing Pearson’s correlations between features, and the visualisation of strong positive and negative correlations divided into 4 classes.

The following tools and libraries were used (open source):Python 3.10.2,Jupyter Notebook 6.4.5,Numpy 1.22.0,Pandas 1.4.1,Matplotlib 3.5.1,Seaborn 0.11.2,Scipy 1.8.0.

Explanation of the methods used:
Inserting missing data. When the set has empty spaces, they can be filled using several strategies, such as inserting the mean, median, or the most common value.Identification of outliers. Outliers are data that do not follow the distribution of other data. Since they are anomalies that should not be modelled, they must be identified and removed. Outliers also have a negative effect on the Pearson’s linear correlation coefficient.Cluster analysis. It is an unsupervised learning method that groups similar data using various algorithms (K-means, Hierarchical Clustering). This method enables cluster analysis, anomaly detection, and dimensionality reduction.Heatmap. A heatmap is a data visualisation technique that shows the magnitude of a phenomenon as a colour in two dimensions. Colour variation can be due to shade or intensity, giving visual clues as to how this phenomenon is clustered and how it changes in space.

In preparing the dataset for analysis, new columns were introduced, the abbreviations of which are shown in Table 2.

The set was divided into two parts:dry friction,fluid friction.

A NaN—Not a Number analysis was performed for the data set, Table 3.

Due to the small number of records (49), selected columns were removed (a large number of NaN values), i.e., Ni, Mo, Mn Co, B, W, V, WC. The remaining columns were completed with the mean values for each column using the SimpleImputer function. Table 4 and Table 5 show the first five elements of both sets with the added mean values.

The descriptive information shown in Table 6 is then presented for the dry friction set, which shows the calculated values of mean, standard deviation, minimum, maximum value, and first, second, and third quartiles. Similar calculations were performed for fluid friction, as shown in Table 7.

In order to demonstrate the correctness of the analyses carried out with regard to the number of tests performed with various modifiers (rare Earth oxides) and their effect on tribological properties, a class chart was prepared. For the tests conducted, the lack of balance between the individual classes may influence the real assessment. Figure 2 shows the number and percentage of the elements per class.

## 3. Results and Discussion

Figure 3 shows the dependence of disc wear on track area for each class.

Figure 4 shows the cumulative disc wear vs. track area. After removing the outliers, it can be seen that the data follow a linear dependence.

In the next step, a cluster analysis was carried out using two methods (K-means, Hierarchical). The analysis showed a high similarity between the two methods used. These are unsupervised classification methods of unsupervised learning.

Figure 5 shows the analyses for dry friction (column on the left) and fluid friction (column on the right).

The first row shows the optimal number of clusters determined using the so-called Elbow method. For both sets, three clusters were determined by the breakpoint. The second row shows the visualisation of the cluster analysis for both cases with the indication of the centroids for all clusters. The next line shows the dendrogram plots for dry and fluid friction on the basis of which the optimal number of clusters for the Hierarchical method was calculated.

The two heatmaps shown above are for the dry friction and fluid friction sets (Figure 6 and Figure 7).

Pearson’s correlations show that in the case of dry friction, a very strong negative correlation occurs between C-cof, C-mwd1, and C-ta1 and a very strong positive correlation occurs between vh-ih and ta1-mwd1. Pearson’s correlation coefficient is a measure of linear correlation between two sets of data. In contrast, for fluid friction, a very strong negative correlation occurs between C-cof with a very strong positive correlation between ta2-mwd2. The results are summarised in Table 8. The correlation values are shown in Table 9.

Several strong correlations can be observed from the heatmaps for dry and fluid friction and for the pairwise correlations between these sets. For dry friction, the C index shows a very strong negative correlation with cof, mtd, and ta, whereas for fluid friction, only a very strong negative correlation with cof is observed. A strong negative correlation was found for ih and ym against the indices from mwd and ta for dry friction. In contrast, this correlation was not observed for fluid friction. For both sets, there is a very strong positive correlation between mtd and ta (0.98—for dry friction, 0.93—for fluid friction). Ym is strongly positively correlated with vh and simultaneously strongly negatively correlated with mtd and ta.

To confirm the correlations discussed above, they are presented in the form of relationships between the variables (Figure 8, Figure 9, Figure 10).

The graphs for dry friction show very strong negative correlations (−0.74, −0.80, −0.74):

Figure 8, Figure 9, Figure 10 show the graphs in which the layers are assigned to classes, i.e., non-modified surface (class 1) and modified surfaces (classes 2, 3, 4). The differences between the individual lines corresponding to the classes are visible. Especially for class 4 (Y_2_O_3_), we observe a different slope of the obtained straight lines. Analysis does not reveal a quantitative but a qualitative relationship, indicating the effects of the additives on the parameters determined.

Exploratory data analysis showed dependencies between individual parameters. The proportionality of the carbon content in the welds with rare earth oxides to their mechanical properties was demonstrated. It was confirmed that its increase resulted in an increase in hardness. This is due to the formation of hard carbides in their microstructure and changes in the morphology of padding welds modified with rare Earth metals. The increase in hardness lowered the coefficients of friction “cof”, which directly translated into the reduction of the surface of the track wear areas and the wear track depths. A slightly different nature of wear was demonstrated for dry friction and fluid friction. It was shown that despite the increase in hardness and the decrease in the friction coefficient, the remaining parameters in the form of the wear track area and depth were not so strongly correlated in the case of fluid friction. SiO_2_ particles were used in the latter case, which changed the nature of the wear into tribocorrosion wear. The analyses confirmed both the complexity of the effect that parameters such as chemical composition or test environment have on wear processes, and the different nature of relationships between the parameters.

## 4. Conclusions

The authors showed that it is reasonable to use exploratory data analysis for the evaluation of the properties of RE oxide modified pad welds. In order to optimise the results, it is advisable to perform a number of additional experiments to expand the database. The expanded database will be used to predict the optimal parameters (chemical composition, percentage of rare Earth elements, padding method, etc.) while maintaining the best wear resistance. The results obtained so far indicate dependency between geometry indices and mechanical properties (friction coefficient, instrumental hardness).

Exploratory data analysis allowed determining directly proportional (positive correlations) and inversely proportional (negative correlations) dependencies. A significant dependency was observed for the “C” index, i.e., the effect of carbon on hardness, friction coefficient, and geometry indices (track area, track depth), which may result from the change in the pad weld morphology after introducing RE oxides. This issue will be addressed by the authors in their next work.

The study and analyses demonstrate that exploratory methods capture non-trivial dependencies of the obtained results.

The analysis of the results of tribological tests is the basis for the development of new techniques of forming surface layers and coatings with increased wear resistance. The wear indicators analysed in this study indicate further direction of surface modification research. The EDA results contributed to the knowledge on the influence of surface modification with rare Earth oxides on wear resistance. The study presented in this article offers a guidance to continue work on the technology of the surfacing process, selection of modifiers, and welding materials for surface layers.

## Figures and Tables

**Figure 1 materials-15-02032-f001:**
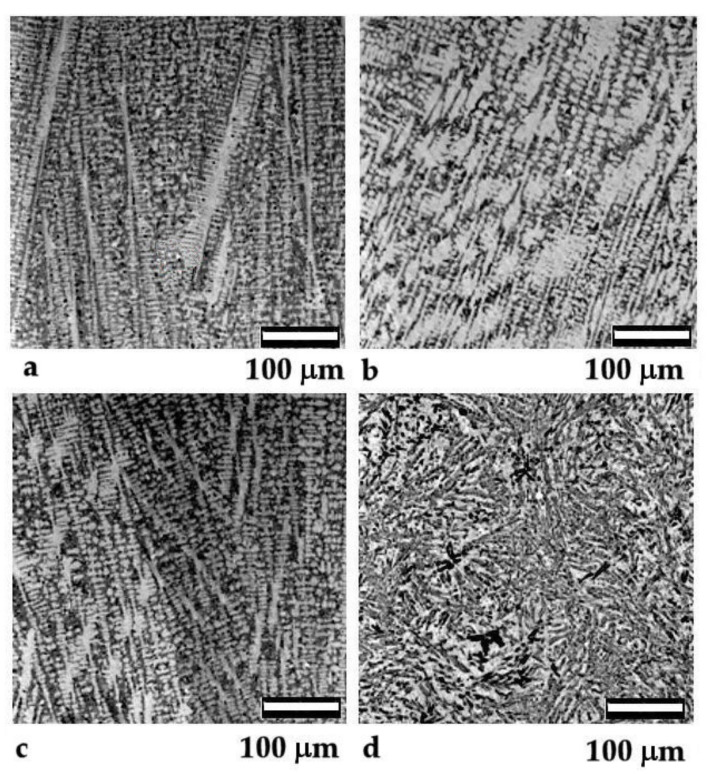
Microstructure of the Ni-based pad weld on the S355 steel: (**a**) non-modified and modified: (**b**) CeO_2_, (**c**) La_2_O_3_, (**d**) Y_2_O_3_.

**Figure 2 materials-15-02032-f002:**
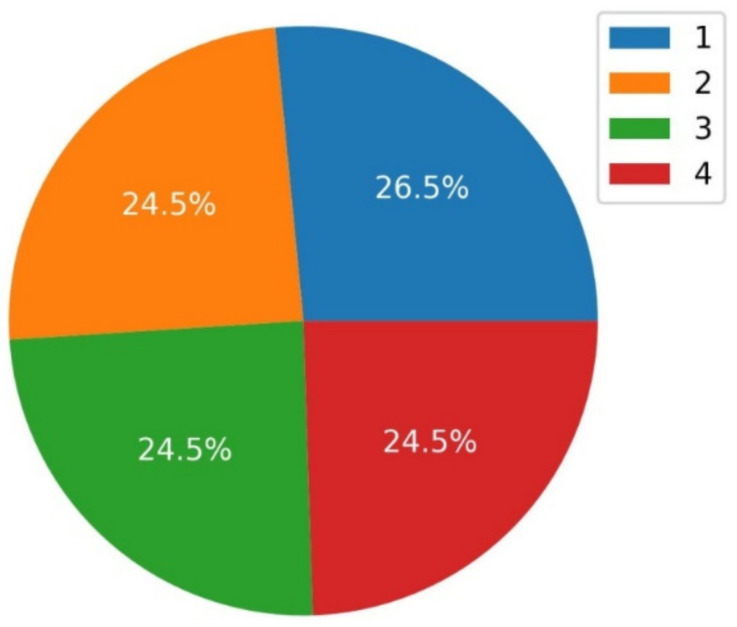
Elements in each class.

**Figure 3 materials-15-02032-f003:**
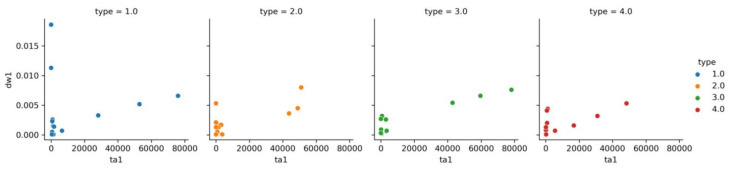
Dependence of dw1 on ta1 separately for each class.

**Figure 4 materials-15-02032-f004:**
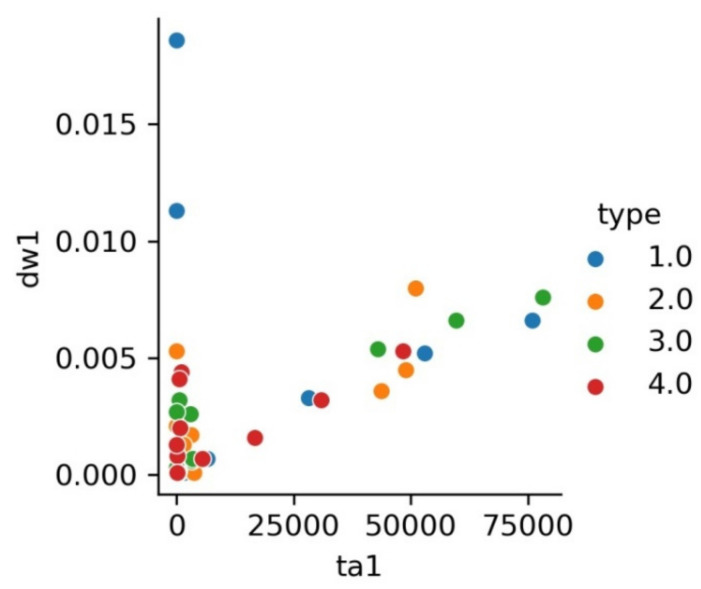
Dependence of dw1 on ta1 by class.

**Figure 5 materials-15-02032-f005:**
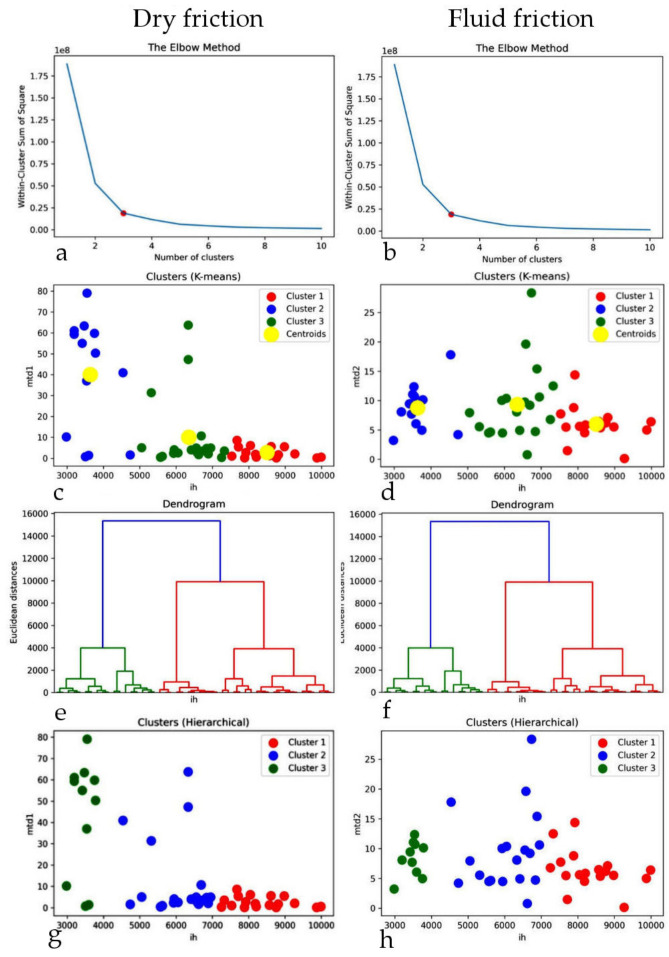
Cluster analysis for dry and fluid friction. Determination of optimal number of clusters using: (**a**,**b**) Elbow method; (**c**,**d**) clusters with centroids determined by K-means algorithm; (**e**,**f**) determination of optimal number of clusters for Hierarchical method using dendrograms; (**g**,**h**) clusters determined by Hierarchical algorithm.

**Figure 6 materials-15-02032-f006:**
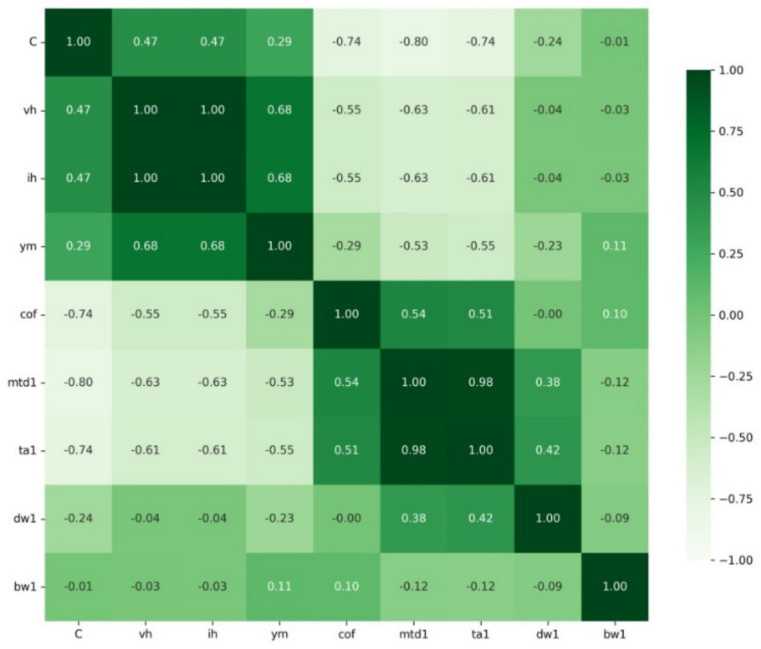
Heatmap for dry friction.

**Figure 7 materials-15-02032-f007:**
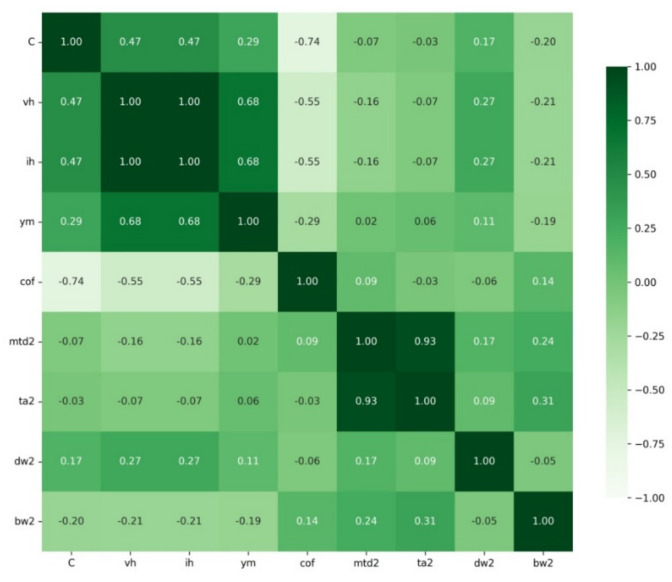
Heatmap for fluid friction.

**Figure 8 materials-15-02032-f008:**
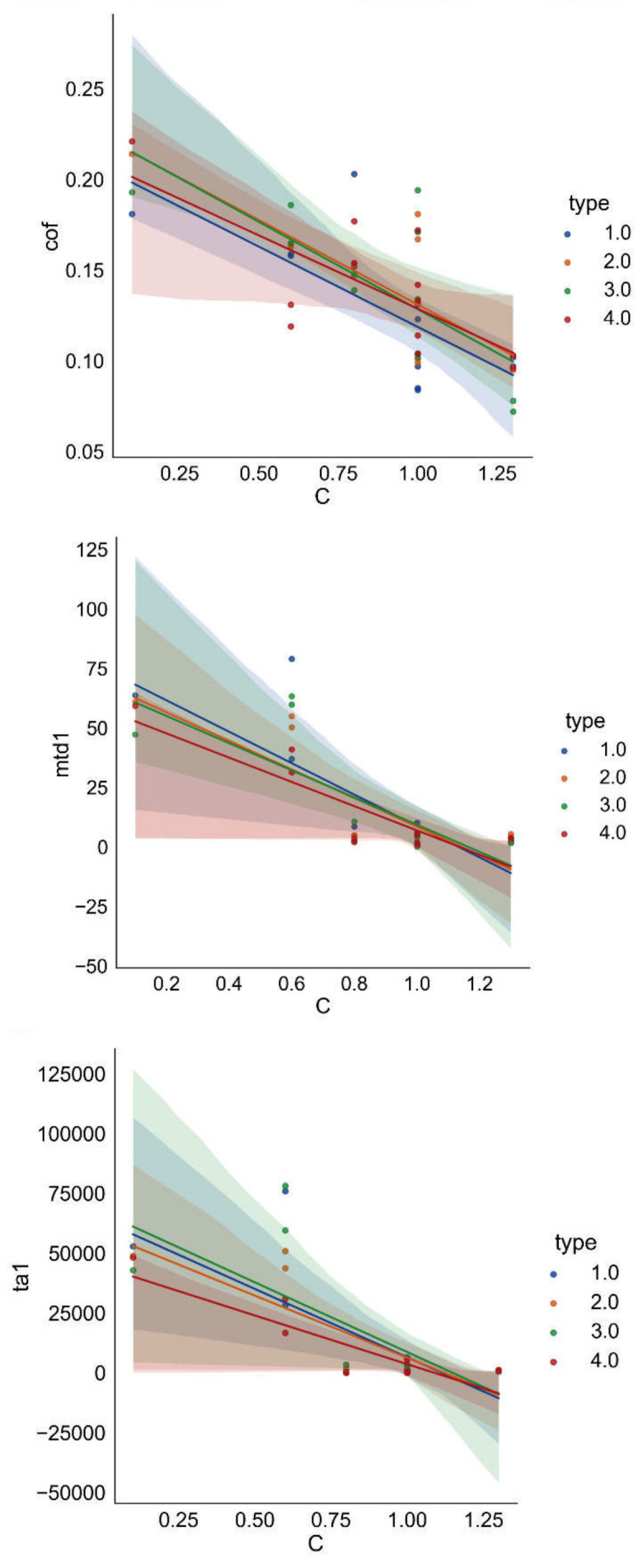
Relationships between the indices and carbon content.

**Figure 9 materials-15-02032-f009:**
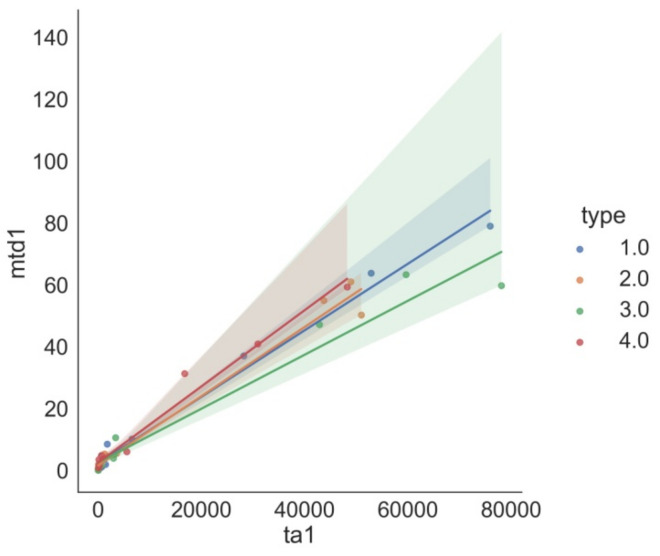
Maximum track depth versus track area—dry friction (positive correlation (0.98)).

**Figure 10 materials-15-02032-f010:**
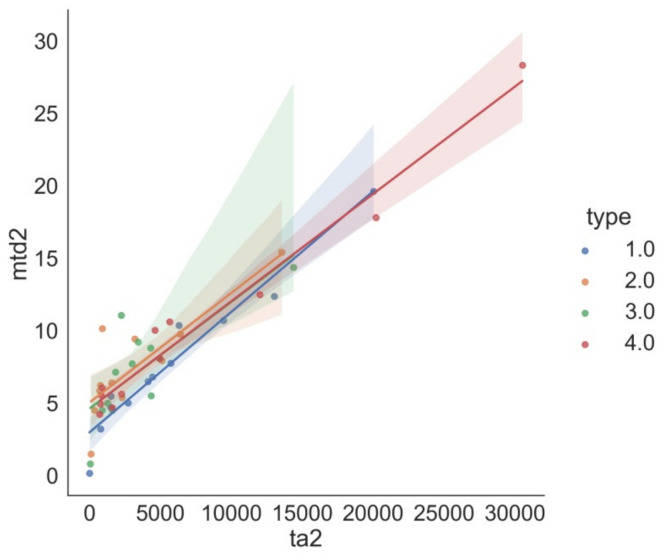
Maximum track depth versus track area—fluid friction (very strong positive correlation (0.93)).

**Table 1 materials-15-02032-t001:** Compilation of sample test results.

			Tribological Tests				Hardness *			Scratch Test *	
			Dry Friction		Friction with Lubricant		Vickershardness (HV)	Instrumental Hardness H_IT_ (MPa)	Young's Moudulus	Coefficient of Friction	Maximum Penetration Depth of the Indenter (μm)
Sample Designation	RE Oxygen	Rare Earth Oxides % mas	Coefficient of Friction	Linear Wear (μm)	Coefficient of Friction	Linear Wear (μm)			(GPa)		
1	-	0	0.06	39.16	-	-	-	-	-	0.181	−27.677
2	CeO_2_	2	0.55	47.24	-	-	301.3	3192.5	207.4	0.214	−13.918
3	La_2_O_3_	2	0.48	33.54	-	-	-	-	-	0.193	−16,982
4	Y_2_O_3_	2	0.57	46.71	-	-	301.7	3196.4	194.1	0.221	−19.903
3-1	-	0	0.78	55.61	0.35	124.26	333.9	3537.6	184.1	0.158	−7.283
3-2	CeO_2_	2	0.82	71.74	0.55	45.45	322.4	3415.9	171.9	0.163	−5.659
3-3	La_2_O_3_	2	0.54	64.1	0.6	57.14	327.7	3471.9	186.5	0.186	−9.281
3-4	Y_2_O_3_	2	0.79	71.07	0.18	96.74	428.2	4536.7	205	0.131	−6.557
4-1	-	0	0.67	55.8	0.5	60.97	335	3549.8	181.2	0.159	−7.46
4-2	CeO_2_	2	0.38	97.73	0.62	64.03	356.7	3779.6	189.5	0.164	−9.219
4-3	La_2_O_3_	2	0.27	55.68	0.52	59.47	354	3750.9	184.7	0.165	−9.338
4-4	Y_2_O_3_	2	0.69	54.15	0.5	97.26	501.4	5312.6	214.5	0.119	−5.899

**Table 2 materials-15-02032-t002:** Description of abbreviations.

Abbrev.	Description	Unit	Test Type
cof1	Coefficient of friction (dry friction)		Tribological tests
lw1	Linear wear (dry friction)	(μm)	Tribological tests
cof2	Coefficient of friction (lubricated friction)		Tribological tests
lw2	Linear wear (lubricated friction)	(μm)	Tribological tests
vh	Vickers hardness	(HV)	Hardness
ih	Instrumental hardness	(MPa)	Hardness
ym	Young’s modulus	(GPa)	Hardness
cof	Coefficient of friction		Scratch test
mpdi	Maximum penetration depth of the indenter	(μm)	Scratch test
mtd1	Maximum track depth (dry friction)	(μm)	Leica
ta1	Track area (dry friction)	(μm^2^)	Leica
mwd2	Maximum track depth (lubricated friction)	(μm)	Leica
ta2	Track area (lubricated friction)	(μm^2^)	Leica
dw1	Disc wear (dry friction)	(g)	Mass loss
bw1	Ball wear (dry friction)	(g)	Mass loss
dw2	Disc wear (lubricated friction)	(g)	Mass loss
bw2	Ball wear (lubricated friction)	(g)	Mass loss
type	1—without additives,		
	2—with an addition of CeO_2_,		
	3—with an addition of La_2_O_3_		
	4—with an addition of Y_2_O_3_		

**Table 3 materials-15-02032-t003:** Analysis of NaN values, before and after the removal of selected columns.

NaN Values before Removal		NaN Value after Removal	
C	0		
Si	4		
Cr	0		
Ni	21		
Mo	24		
Mn	45		
Fe	8		
Co	41		
B	25	C	0
W	20	Si	4
V	28	Cr	0
WC	41	Fe	0
cof1	0	cof1	0
lw1	0	lw1	0
cof2	4	cof2	4
lw2	4	lw2	4
vh	2	vh	2
ih	2	ih	2
ym	2	ym	2
cof	0	cof	0
mpdi	0	mpdi	0
mwd1	0	mwd1	0
wa1	0	wa1	0
mwd2	4	mwd2	4
wa2	4	wa2	4
dw1	0	dw1	0
bw1	0	bw1	0
dw2	4	dw2	4
bw2	4	bw2	4
type	0	type	0
dtype:	int64	dtype:	int64

**Table 4 materials-15-02032-t004:** Five elements of the set for dry friction with the added means.

No.	C	Si	Cr	Fe	cof1	lw1	vh	ih	ym	cof	mpdi	mtd1	ta1	dw1	bw1	Type
1	0.1	2.1	18.0	62.9	0.06	39.16	598	6330	214	0.181	−27,667	63.79	52,888	0.0052	0.0003	1.0
2	0.1	2.1	18.0	62.9	0.55	47.24	301	3193	207	0.214	−13,918	61.05	48,955	0.0045	0.0001	2.0
3	0.1	2.1	18.0	62.9	0.48	33.54	598	6330	214	0.193	−16,982	47.23	42,889	0.0054	0.0000	3.0
4	0.1	2.1	18.0	62.9	0.57	46.71	301	3196	194	0.221	−19,903	59.33	48,247	0.0053	0.0000	4.0
5	0.6	3.8	11.0	3.0	0.78	55.61	334	3538	184	0.158	−7,283	37.07	28,214	0.0033	0.0001	1.0

**Table 5 materials-15-02032-t005:** Five elements of the set for fluid friction with the added means.

No.	C	Si	Cr	Fe	cof2	lw1	vh	ih	ym	cof	mpdi	mtd2	ta2	dw2	bw2	Type
1	0.1	2.1	18.0	62.9	0.38	58.16	598	6330	214	0.181	−27,667	8.09	4939	0.0019	0.0008	1.0
2	0.1	2.1	18.0	62.9	0.38	58.16	301	3193	207	0.214	−13,918	8.09	4939	0.0019	0.0008	2.0
3	0.1	2.1	18.0	62.9	0.38	58.16	598	6330	214	0.193	−16,982	8.09	4939	0.0019	0.0008	3.0
4	0.1	2.1	18.0	62.9	038	58.16	301	3196	194	0.221	−19,903	8.09	4939	0.0019	0.0008	4.0
5	0.6	3.8	11.0	3.0	0.35	124.26	334	3538	184	0.158	−7.283	12.37	13,033	0.0090	0.0140	1.0

**Table 6 materials-15-02032-t006:** Basic information for the set—dry friction.

	C	vh	ih	ym	cof	mtd1	ta1	dw1	bw1
count	49	49	49	49	49	49	49	49	49
mean	0.877	597	6330	214	0.138	15.60	12,675.0	0.00289	0.00024
std	0.317	187	1982	20	0.039	23.13	22,171.4	0.00338	0.00018
min	0.100	281	2980	172	0.072	0.18	8.7	0.0010	0.00000
25%	0.800	447	4733	202	0.102	1.58	177.5	0.0007	0.00010
50%	1.000	625	6579	214	0.134	3.95	1093.0	0.0017	0.00020
75%	1.000	744	7885	225	0.165	10.66	78,109.0	0.0041	0.00030
max	1.300	942	9985	257	0.221	70.05	78,109.0	0.01860	0.00080

**Table 7 materials-15-02032-t007:** Basic information for the set—fluid friction.

	C	vh	ih	ym	cof	mtd2	ta2	dw2	bw2
count	49	49	49	49	49	49	49	49	49
mean	0.877	597	6330	214	0.138	8.09	4938.5	0.00194	0.00082
std	0.317	187	1982	20	0.039	4.90	6094.1	0.00184	0.00195
min	0.100	281	2980	172	0.072	0.16	4.6	0.00000	0.00000
25%	0.800	447	4733	202	0.102	5.02	912.8	0.00070	0.00030
50%	1.000	625	6579	214	0.134	7.14	3008.0	0.00130	0.00050
75%	1.000	744	7885	225	0.165	10.03	5109.0	0.00220	0.01400
max	1.300	942	9985	257	0.221	28.33	30,517.0	0.00760	0.01400

**Table 8 materials-15-02032-t008:** Significant correlations between the results.

Correlation	Dry Friction				Fluid Friction			
	Positive		Negative		Positive		Negative	
	Strong	V.Strong	Strong	V.Strong	Strong	V.Strong	Strong	V.Strong
	0.5–0.7	0.7–1	−0.5–0.7	−0.7–1	0.5–0.7	0.7–1	−0.5–0.7	−0.7–1
**C**				Cof, mwd, wa				Cof
**vh**	Ym	Ih	Cof, mtd, ta		Ym	Ih	Cof	
**ih**	Ym	Vh	Cof, mtd, wa		Ym	Vh	Cof	
**ym**	Vh, ih		mtd, ta		Vh, ih			
**cof**	mtd, ta		Vh, ih	C			Vh, ih	C
**mtd**	Cof	ta	Vh, ih, ym	C		ta		
**ta**	Cof	mtd	Vh, ih, ym	C		mtd		

**Table 9 materials-15-02032-t009:** Strength of association.

Correlation	Negative	Positive
Very strong	−0.7 to −1	0.7 to 1
Strong	−0.5 to −0.7	0.5 to 0.7

## Data Availability

Data sharing not applicable, all the data created for this study are already displayed in the article.

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
