# Peer review of "Exploratory Data Analysis for the Evaluation of Tribological Properties of Wear-Resistant Surface Layers Modified with Rare-Earth Metals"

_materials, 2022, doi:10.3390/ma15062032_

Round 1

Reviewer 1 Report

Dear authors,

Find my comments

Author Response

  1. How the indirect analysis solves the wear issues?
    2. The abstract lacks full information
    3. And what was the result in your exploratory analysis in your abstract?
    4. Even you did not mention if you applied confusion matrix in your abstract (as a standard,
    keywords must me mentioned at least once your abstract like heatmap, confusion matrix etc)
    5. Line 32 ...Recent years have seen... subject verb dialogue...who have seen it.... language
    problem.
    6. What L indicates in your reference .... follow template
    7. Y2O3 or Y2O3 and many others//written carelessly
    8. Be consistent in use of bullets ...one you started small letters line 75// and the other capital
    letter line//91 etc
    9. How did you perform machine learning with such small data?
    10. It is not clear which machine learning algorithm you performer

Ad.1.

The analysis of the results of tribological tests is the basis for the development of new techniques of forming surface layers and coatings with increased wear resistance. The wear indicators analysed in this study indicate further direction of surface modification research. The EDA results contributed to the knowledge on the influence of surface modification with rare earth oxides on wear resistance. The study presented in this article offers a guidance to continue work on the technology of the surfacing process, selection of modifiers and welding materials for surface layers.

The text has been added.

Ad.2.

The Abstract has been reviewed.

Ad.3.

The Abstract has been reviewed and corrected.

Ad. 4.

The confusion matrix was not used in this study as it is one of the measurement methods for the performance of classification model. The publication did not include modelling due to the small amount of data. The abstract  has been reviewed to include the information suggested by the Reviewer.

Ad. 5.

The Reviewer’s suggestion has been implemented. The article was translated by a professional translator

Ad.6.

The error has been corrected.

Ad.7.

The writing has been improved.

Ad.8.

The use of bullets has been improved.

Ad.9.

Machine learning was not performed because of insufficient amount of data.

Ad.10.

Due to the fact that machine learning was not performed, no algorithms were indicated. 

Reviewer 2 Report

The paper ”Exploratory data analysis for the evaluation of tribological properties of wear-resistant surface layers modified with rare-earth metals” is suitable for publication in Materials with some addition of other analysis.

In my opinion, the mechanical results should be correlated with some microstructural aspects regarding the behavior of the layers. So, some microstructural images should be added. 

Author Response

The text has been supplemented with the information suggested by the Reviewer.

Reviewer 3 Report

The authors have studied the effect of rare earth oxides on the wear resistance of surface layers applied on cast parts for structural components of machinery and equipment and found that addition of cerium, yttrium and lanthanum oxides while pad welding improved weld tribological properties. Using hardness, linear wear, mass loss and other parameters of the weld for exploratory data analysis, they revealed the complexity of analysis when testing was conducted under different conditions with various modifiers. The manuscript is interesting, but some deficiencies need to be corrected to make it acceptable for publication.

(1) The values of applied load in the cases of tribological and hardness tests should be given in the section “Materials and Methods”.

(2) It is suggested that the author would use a term “characteristics of wear resistance” or “wear resistance characteristics” or “wear resistance” instead of an improper term “anti-wear characteristics/properties”.

(3) Line 28: please check the writing of the word “”oxide”.

(4) In the section “Conclusions” after the sentence “The results of the experiments performed so far demonstrated the relationships between the indices of the geometric structure and mechanical properties (the friction coefficient and instrumental hardness)” conclusion should be drawn on other parameters used in this study for which no correlation was found.

(5) The publication year is absent for References 13, 14, and 24.

(6) References for 2018-2022 should be added.

Author Response

(1) The values of applied load in the cases of tribological and hardness tests should be given in the section “Materials and Methods”.

(2) It is suggested that the author would use a term “characteristics of wear resistance” or “wear resistance characteristics” or “wear resistance” instead of an improper term “anti-wear characteristics/properties”.

(3) Line 28: please check the writing of the word “”oxide”.

(4) In the section “Conclusions” after the sentence “The results of the experiments performed so far demonstrated the relationships between the indices of the geometric structure and mechanical properties (the friction coefficient and instrumental hardness)” conclusion should be drawn on other parameters used in this study for which no correlation was found.

(5) The publication year is absent for References 13, 14, and 24.

(6) References for 2018-2022 should be added.

Ad.1.

The values have been added.

Ad.2.

The incorrect term has been replaced with “wear resistance”.

Ad.3.

The spelling has been checked and corrected.

Ad.4.

The suggested information has been added.

Ad.5.

The publication year has been added.

Ad.6.

Recent references have been added.

Reviewer 4 Report

In this paper, EDA method is used to evaluate the impact of rare earth surface modification on anti-wear performance, which is a good innovative application in tribological data analysis. However, in the article, there is no clear test conditions and test methods, and the test data is not credible.

Despite the use of EDA method, the effect of rare earth oxides on surface wear resistance has not been evaluated from the analysis process or final conclusion.

The figures and tables in the manuscript are not standardized

Author Response

(The authors gave the same response as above.)

Reviewer 5 Report

The study by Malinowski et al. attempts for complex statistical research of data related to tribological performance of RE-oxides-modified surface layers by so-called exploratory data analysis. Although the manuscript contains various data obtained by a number of testing and analysing approaches, it is very difficult to follow the line of their interpretation. The structure of the manuscript is inappropriate. The section 2 is very poor on details concerning both the experimental testing (missing details about experimental devices and testing conditions) and the design of exploratory data analysis. Huge part of the section 3 should be rather placed into section 2, however, it would be still insufficient to identify the meanings of individual terms of the analysis. Simply, nomenclature of EDA is missing which makes the article highly non-understandable. Nevertheless, the data analyses shown in present manuscript do not really bring exploratory results (conclusions) as the authors admit themselves that the dataset of the analysed data was too small. Taking into account all these drawbacks, I do not recommend the manuscript for publishing in journal Materials. I am sorry...

Author Response

(The authors gave the same response as above.)

Round 2

Reviewer 1 Report

Here are some my comments
1. First of all, I cannot see the changes from the previous one as there is no highlights or track changes which I THOUGHT the authors did carelessly
2. The abstract has a serious flow problem which the normal article follows that is introduction→purpose→methods→results→conclusion. In addition, if I want to read the article I cannot read all (as a reader), I must get full information which I did not get from it. The other thing is the abstract is not quantitatively expressed as it is experimental
3. The other thing is the title is not simple and explanatory, the word Exploratory data analysis is confusing one which I cannot get the hint in the whole text
4. The introduction section has flow problems: a) does not attract the Reader's Attention b) does not state the focused topic
5. Referencing problems (have a look reference 19-22 and 23-25 (the gap looks manual)
6. In methods section has also flow problems; the authors started “materials” and the stated about “methods” which make confusion to the readers
7. Have a look bullet using techniques which is written carelessly
8. Have a look subtitle 2.3 is confusing (the author start to include data acquisition and data wrangling in ED and then they said the first two are from study and experiments; much confusing
9. It is not clearly indicated where exactly the tools are used (python, pandas etc)
10. Figure 8, 9 and 10 are not clearly described and labelled
11. Over all the article has serious flow problems and much confusing with full of jargon words; that is my feeling

Author Response

Thank you very much for your valuable comments. We have made corrections in line with the Reviewer’s suggestions.

Ad.1.

We have prepared the manuscript without the track changes option, which, we agree, may make it difficult to re-evaluate the article. We apologize for this mistake. The revised version of the article we are currently posting includes the option of tracking changes made according to the comments of all the Reviewers in Round 2.

Ad.2.

The Abstract has been rewritten according to the Reviewer’s suggestion to follow the “introduction→purpose→methods→results→conclusion” pattern. We think, however, that since all the necessary details are contained in the main text, it is not necessary to include them in the abstract.

Ad.3.

The Exploratory Data Analysis EDA is a kind of descriptive analysis used to evaluate the data obtained. It also allows creating diagrams that help to better understand the data through visualisation.

Ad.4.

The Introduction section has been supplemented with the information on the purpose of the study.

Ad.5.

The literature items mentioned have been rewritten to have the form 19, 20, 21, 22 and 23, 24, 25.

Ad.6.

The authors introduce the reader to the issues of the modification, application, and tribological testing of pad welded layers. However, these issues are not the subject of this paper. Section 2.3. Exploratory Data Analysis, which is the main topic of the article, has been described in detail.

Ad.7.

The introductory part of the “Materials” section has been rewritten.

Ad.8.

The introductory part of subsection 2.3 has been rewritten (shortened to the items used) for clarity.

Ad.9.

The tools used in the process of exploratory data analysis were as follows: Python, Jupyter Notebook, Numpy, Pandas, Matplotlib, Seaborn,Scipy.

Python is a programming language that lets you work quickly and integrate systems more effectively. The Jupyter Notebook is the original web application for creating and sharing computational documents. It offers a simple, streamlined, document-centric experience.

Numpy is the fundamental package for scientific computing with Python Pandas is a fast, powerful, flexible, and easy to use open source data analysis and manipulation tool, built on top of the Python programming language.

Matplotlib is a comprehensive library for creating static, animated, and interactive visualizations in Python. Matplotlib makes easy things easy and hard things possible.

Seaborn is a Python data visualization library based on matplotlib. It provides a high-level interface for drawing attractive and informative statistical graphics.

SciPy provides algorithms for optimization, integration, interpolation, eigenvalue problems, algebraic equations, differential equations, statistics, and many other classes of problems.

The publication does not concern the methodology of data analysis with the specification of tools, libraries, methods and techniques used, but it concerns the use of exploratory data analysis methods in tribology.

 Ad.10

The figures have been corrected.

Ad.11

The paper has been rewritten according to the suggestions received in Round 2.

Reviewer 2 Report

In my opinion, the paper is ok now!

It can be accepted for publication.

Author Response

Thank you for your comments. We are re-submitting the manuscript for the review.

Reviewer 4 Report

The manuscript still has the serious flaws proposed in the previous review

Author Response

Thank you very much for your comments. In the meantime, the article was sent without an option to track changes, which might make it difficult to re-evaluate, for which the authors apologize.

The information on the tests carried out has been supplemented.

In addition, the authors have addressed the results obtained in connection with the classification into classes. The caption for Figure 10 has been supplemented with the relevant information.

“Figures 8 to 10 show the graphs in which the layers are assigned to classes, i.e. non-modified surface (class 1) and modified surfaces (classes 2, 3, 4). The differences between the individual lines corresponding to the classes are visible. Especially for class 4 (Y2O3), we observe a different slope of the obtained straight lines. Analysis does not reveal a quantitative but a qualitative relationship, indicating the effects of the additives on the parameters determined.”

The figures and tables have been standardized.

Reviewer 5 Report

The manuscript has been improved, however, I cannot be completely satisfied with performed changes as they were not highlighted clearly. Anyhow, please, consider the additional comments:

1.) Figure 2. - Could it be rather presented as Table?  I consider, the interpretation in the form of "Table 3" would be more appropriate than your current interpretation in the form of "Figure 2".

2.) In current Tables 3 and 4, please, remove arow-cursors.

3.) Table 7 - Could it be rather presented as Figure?  I consider, the interpretation in the form of "Figure 6" would be more appropriate than your current interpretation in the form of "Table 7".  Then figure captions should be defined thoroughly for all sub-figures within newly designed Figure 6.

Accordingly, the rest of the Tables and Figures should be re-numbered.

Author Response

We appreciate your valuable comments in both reviews. All comments have been taken into account. We are sending the manuscript with the option of tracking changes.

Ad.1.

Figure 2 has been replaced with a table.

Ad.2.

The cursors have been removed.

Ad.3.

Table 7 has been replaced with a drawing. The numbering of figures and tables in the text has been corrected.

Round 3

Reviewer 1 Report

Ok

Author Response

The authors are grateful to all the Reviewers  for meticulous reviews and all comments.

The final amended version of the article (without the tracking option) is being submitted. The corrections include:

- conversion of the tables to the "three wire table" format

- convertion of  tables 4-7 (screenshots to word tables)

- correction of Figure 1 and Tables 1 and 3 - captions

The authors have made efforts to ensure that the final form is suitable for approval for publication.

Reviewer 4 Report

In this paper, the quantitative evaluation of tribological properties by EDA method is a meaningful exploration However, after two revisions, the manuscript still has some errors and problems, which can only be published publicly

1) The test substrate is inconsistent with the test purpose. In the abstract, the purpose of the test is to solve the problem of wear resistance on the surface of the casting. In the 110 line, the test substrate is S355 steel. In the test results (Fig. 1), the test substrate is changed into Ni based material

2) All tables are not standardized and should be changed to three wire table

3) The manuscript does not clearly state the test conditions of the data in Table 1, especially the RE oxide content in the second column is different, and it is also proportioned with different powders The third column is blank

4) The title of Table 3 is not clear, the first and third columns in the table can be combined, and the source of test data is unknown. Is it related to table 1?

5) Tables 4 to 7 are not standardized and cannot be represented by screenshots

Author Response

The Authors are grateful for the detailed analysis and all the comments.

Ad.1.

The error in the caption to Figure 1 has been corrected.

Figure 1. Microstructure of the Ni – based pad weld on the S355 steel: a) non-modified and modified: b) CeO2, c) La2O3, d) Y2O3.”

Ad.2.

All tables in the manuscript have been converted to three wire table format.

Ad.3.

Table 3 includes some examples of the results obtained and analysed. The Excel spreadsheet contains significantly more records. For the analysis, padding welds with one content of RE oxides were selected in comparison with various padding welds. Column 3 ("blank") contains the content (mass%) of RE oxides.

“Table 1. Compiled examples of test results “

Ad.4.

Table 3 has been corrected.

"Table 3. Analysis of NaN values, before (column a) and after  (column b) the removal of selected columns."

Ad.5.

Tables 4 to 7 have been converted to standardised forms.

Reviewer 5 Report

The authors addressed the comments of the reviever.

Author Response

(The authors gave the same response as above.)
